# Hemolytic Activity of Nanoparticles as a Marker of Their Hemocompatibility

**DOI:** 10.3390/mi13122091

**Published:** 2022-11-27

**Authors:** Saul Yedgar, Gregory Barshtein, Alexander Gural

**Affiliations:** 1Department of Biochemistry, The Faculty of Medicine, Hebrew University, Jerusalem 91120, Israel; 2Blood Bank, Hadassah-Hebrew University Medical Center, Jerusalem 91120, Israel

**Keywords:** nanomaterials, nanoparticles, red blood cells, hemocompatibility, nanotoxicity, hemolytic activity

## Abstract

The potential use of nanomaterials in medicine offers opportunities for novel therapeutic approaches to treating complex disorders. For that reason, a new branch of science, named nanotoxicology, which aims to study the dangerous effects of nanomaterials on human health and on the environment, has recently emerged. However, the toxicity and risk associated with nanomaterials are unclear or not completely understood. The development of an adequate experimental strategy for assessing the toxicity of nanomaterials may include a rapid/express method that will reliably, quickly, and cheaply make an initial assessment. One possibility is the characterization of the hemocompatibility of nanomaterials, which includes their hemolytic activity as a marker. In this review, we consider various factors affecting the hemolytic activity of nanomaterials and draw the reader’s attention to the fact that the formation of a protein corona around a nanoparticle can significantly change its interaction with the red cell. This leads us to suggest that the nanomaterial hemolytic activity in the buffer does not reflect the situation in the blood plasma. As a recommendation, we propose studying the hemocompatibility of nanomaterials under more physiologically relevant conditions, in the presence of plasma proteins in the medium and under mechanical stress.

## 1. Introduction

Engineered man-made nanomaterials have several applications in the field of biomedicine for diagnosis [1], drug delivery [2], and therapeutics [3]. The International Organization for Standardization defines nanomaterials as structures with a size range from 1 to 100 nm in one, two, or three dimensions [4]. Scanning Electron Microscopy (SEM) and Transmission Electron Microscopy (TEM) [5,6] are the primary tools for the visualization of nanomaterial shapes (as illustrated in Figure 1). An extensive library of images of various nanomaterials has been collected [7,8,9,10,11].

Nanomaterials are drawing increasing interest from many branches of medical practices and research [6]. Their use in medical devices or as drug carriers offers opportunities for novel therapeutic approaches to treat complex disorders such as malignant, inflammatory, and neurodegenerative diseases [12,13,14].

Humans may be exposed to nanomaterials through inhalation (respiratory tract), skin contact, ingestion, or intravenous (IV) injection. The tiny size of nanomaterials allows for them to pass more easily through cell membranes [15,16]. Moreover, some nanomaterials are readily distributed throughout the body, where they are deposited in the mitochondria of the target organs and may trigger tissue injury [15]. Possible pathways for nanoparticle uptake and intracellular transport routes have been extensively discussed in the literature, and several recent reviews are hereby recommended [17,18].

Despite the advantages offered by nanotechnology, the potential risk of intended and unintended human exposure to nanomaterials is increasing as nanotechnology develops. Novel nanomaterials are currently widely used without thoroughly assessing their potential health risks. The knowledge regarding their toxic potential is still limited, without appropriate regulatory measures being implemented [4,20,21].

Early studies on asbestos and man-made nanomaterials, such as diesel exhausts, have shown that they can accumulate in the human body, especially after daily exposure, such as in occupational settings. Long-term and short-term toxicity to humans and animals caused by nanomaterials has already become a serious concern. Therefore, a new branch of science, named nanotoxicology [22,23], has emerged, aiming to study nanomaterials’ hazardous effects on human health and on the environment.

In many cases, novel nanoparticles (NPs) are widely manipulated without thoroughly assessing their potential health risks. The broad range of composition and physicochemical properties of NPs (colloidal stability, purity, inertness, size, shape, charge, etc.) make them ubiquitous and determine their interactions with other biological materials and the extent of their toxicity [24,25]. As with regular particles, the NP surface forms the point of contact with cells. Therefore, surface area and surface chemistry are essential determinants of NPs toxicity [26]. The geometry of NPs, determining their effective surface area, affects not only the interaction between NPs and plasma proteins but also the mechanism and degree of cellular absorption and, consequently, the potential toxicity of NPs [27].

Several approaches can be used to assess NP toxicity; these include epidemiological studies, human clinical studies, animal models, and in vitro models [28,29,30,31,32,33].

Whatever their use, source, and route of exposure (whether oral, respiratory, or dermal), NPs can enter the bloodstream. Several studies have reported that, due to their small size, free NPs can penetrate the alveolar lining [34,35,36], cause inflammatory reactions, and subsequently enter the bloodstream [37]. The circulation then distributes such NPs throughout the body, allowing their penetration into various organs, where they are partially metabolized, excreted, or retained. Moreover, in the bloodstream itself, the NPs interact with various blood cells [27], especially red blood cells (RBCs), the most abundant cellular component in circulation. The exposure of RBCs to NPs leads to various biochemical/biophysical and morphological changes that can significantly affect their functionality [38,39].

Under in vivo conditions (in the bloodstream), the contact between an NP and an RBC occurs in plasma, where all its components (proteins, hormones, vitamins, sugars, and inorganic ions) can affect this interaction. The effect of proteins is the most studied of all the plasma components. It is convincingly documented that the particle’s surface is covered with a corona formed by adsorbed proteins in the plasma [40]. However, most publications on this subject describe NP–RBC interaction occurring in a buffer. Accordingly, this aspect is the focus of the first section of our review. Next, we briefly discuss the process of corona formation around an NP. Several recent reviews [41,42,43] are recommended for a more detailed presentation of this subject. The following section examines the RBC interaction with corona-coated NPs. A separate section discusses the various methods for assessing NP hemotoxicity. In the last section, we outline the directions for further research in this area.

## 2. Interaction of NPs with Red Blood Cells (RBCs) in a Protein-Free Medium

NPs (see Table 1) interact with cells differently than small molecules and are incorporated into the cell by active, energy-dependent processes. Direct NP/RBC contact can cause a change in the state of the cell membrane [44] and, in many cases, disrupts the membrane integrity leading to hemolysis. It has been previously shown that the adsorption of NPs onto the RBC surface can provoke alterations in cell morphology [45,46], the elevation of osmotic fragility [47] and rigidity [48], alterations in cells’ aggregability and adhesion to endothelial cells [49], and membrane vesiculation [50]. The consequences of the NP interaction with a cell are discussed in detail in a recent review by Tian et al. [50].

As has been summarized in several reviews [82,83], RBC hemolysis is the most extensively discussed effect of NPs. The NP hemolytic activity is considered the primary criterion for hemocompatibility [82]. Oberdörster et al. [26] proposed a list of physicochemical characteristics that might be important for understanding the biological activity and toxic properties of NPs.

In particular, the hemolytic activity of nanomaterials has been extensively studied using polystyrene nanoparticles (NPPS) [48,49,54,84]. In a previous study [49], we reported that the NPPS hemolytic activity is a function of their concentration, size, and protein concentration in the medium.

The mechanism of hemolysis induced by NPPS has not yet been defined, but it nevertheless has to be NP adhesion-dependent so that changes in RBC/NP interaction conditions will modulate the level of cell hemolysis. Thus, the destabilization of the RBC membrane [40] by the interaction of NP with the cell lipid bilayer may activate membrane defects [41,42] that cause RBC hemolysis, implying that the attenuation of NP adhesion to RBC can reduce the hemolysis.

Peetia and Labhasetwar [85] observed that plain NPPS induced a decrease in the cell membrane surface pressure, which was inversely proportional to the particle size, indicating a loss of phospholipids from the interface into the bulk. The authors [86] related this to the interaction of the phospholipid hydrophobic chains with hydrophobic NPs, which then mobilize the phospholipid molecules from the interface into the subphase, causing destabilization of the membrane. Moreover, the authors conclude that the modification of the particle surface leads to significant changes in the nature of its interaction with the cell membrane. It was found that double-stranded and single-stranded cationic surfactants on NPs interact differently with model membranes [86]. NPs that exhibit stronger biophysical interactions with the membrane also show greater cellular uptake.

Moreover, the authors conclude that the functionalization of the particle surface leads to significant changes in the nature of its interaction with the membrane [86]. It was found [86] that the di-chained and single-chained cationic surfactants on NPs have different interaction mechanisms with model membranes. Saha et al. [87] found that a linear hemolytic profile with increasing NP surface hydrophobicity is exhibited in the absence of plasma proteins.

The generation of oxidative stress (OS) by NPs is widely discussed in the literature [88,89,90], with convincing data suggesting that it is a common cause of damage to RBCs [91,92,93,94,95,96,97], leading to cell dysfunction [94,96,97,98] and ultimately to hemolysis [99,100]. Several studies have demonstrated the significance of reactive NPs’ surface in ROS generation [90,101]. Free radicals are generated when the oxidants and free radicals are bound to the particle surface. For example, for silica NP (NPSiO_2_), surface-bound radicals such as SiO• and SiO_2_• are responsible for the formation of ROS such as OH• and O_2_• [102].

Special attention was paid to evaluating the undesirable effects of gold and silver NPs (NPAu/NPAg), which are increasingly used in biomedical applications [68,69,72,74,103]. The increased interest in these nanoparticles is associated with their ability to penetrate bacterial cell membranes, change the structure of cell membranes, and even lead to cell death [104]. The effectiveness of NPAg is due to its nanosize, large surface area to volume ratio, and the ability to produce reactive oxygen species and release silver ions [105]. Finally, the generation of ROS and OS by gold and silver NPs leads to cytotoxicity and genotoxicity. [105,106]. Regarding the effect of gold and silver nanoparticles on RBCs [107,108], it was found that their incubation with cells caused significant hemolysis [109].

Interestingly, the interaction between NPAg and a red cell leads not only to a change in its membrane composition but also to an alteration in intracellular hemoglobin properties. Barkur et al. [110] studied the effect of NPAg and NPAu on RBCs using Micro-Raman Spectroscopy and observed spectral modifications, which implicate the deoxygenation of hemoglobin in NP-treated RBCs. The interaction of RBCs with NPs generally adversely affects the hemoglobin’s ability to bind oxygen, with NPAg demonstrating a relatively more substantial adverse effect than NPAu [110]. The authors hypothesized that OS triggered by NPAg caused more profound changes in the RBCs and, consequently, higher spectral variations. Barkur et al. [110] confirmed the two mechanisms involved in metal NP-induced hemoglobin deoxygenation on RBCs: the adherence of NP to the RBC membrane and OS generation. Perevedentseva et al. [61] also used Raman Spectroscopy to study the effect of NPTiO_2_ on the hemoglobin oxygenation state in the RBC cytoplasm. The authors postulated that the adsorption of NPTiO_2_ onto the cell surface leads to the partial deoxygenation of hemoglobin [61].

## 3. RBC as Carriers of Nanoparticles

Since RBCs are the most abundant cellular component in circulation, RBC-based drug delivery systems (DDSs) [111] have been the subject of extensive research in the last decades [112,113,114,115,116]. “Hitchhiking with RBCs” is a drug-delivery method that can increase drug concentration in target organs by orders of magnitude [117]. Accordingly, a new class of delivery systems [71,118,119,120] has been developed, consisting of human RBCs bearing NPs loaded with therapeutic agents [118]. In addition, some groups have taken a new approach to increase the circulation time of NPs by forming an RBC-NP complex, which reduces the rate of NPs’ removal from the bloodstream [121,122]. Since the attachment of NP to RBC leads to a significant change in a wide range of cell properties, a thorough study of the RBC-NP complex behavior in vitro and in vivo is necessary. In particular, it is essential to assess the sensitivity of the RBC-NP complex to oxidative, mechanical, and osmotic stresses [47,48].

Several studies have demonstrated the applicability of this approach in nanomedicine [111,123,124]. However, the effect of modified cells on the behavior of native RBCs has been little studied. In this regard, of particular interest is the work of Barshtein et al. [38], which examined the effect of RBC-NPPS on the aggregation of RBCs and their adhesion to endothelial cells (EC). Red cells were incubated with NPPS, washed, and added to a suspension of untreated RBCs at varying concentrations. The RBC-NPPS complexes induced red cell aggregates (in PBS) and markedly elevated RBC adhesion to EC. These effects were augmented by (a) increasing the concentration of RBC-NPPS and (b) decreasing the NPPS size. This implies that the RBC-NP complex can induce strong interaction with native RBCs and form large and robust aggregates with native red cells [38,39,125], as well as enhance RBC/EC interaction [58,99]. Han et al. [39] discussed the mechanism of RBC aggregation that was modulated by hydroxyapatite NPs and concluded that NP-induced RBCs aggregation could be attributed to the bridging force via the surfaces of NPs and RBCs. The authors consider two alternative RBC aggregation models proposed to describe RBCs aggregation in a medium containing macromolecule and suggest the bridging model [126,127] as a preferred one.

## 4. Corona Formation

In the blood, a layer of plasma components is adsorbed onto the NP surface, modifies its properties, and imparts it with a new identity [128,129]. Therefore, under physiological conditions, RBCs do not directly interact with the NP surface, but rather with plasma proteins bound to the particle with varying strengths [130,131,132], named “corona” [129,133,134,135]. For a single-protein solution, it has been shown that the protein binds to the NP with micromolar affinity, depending on surface properties [134,136,137]. However, when NP is suspended in plasma (which contains numerous types of proteins), proteins that first adsorb to its surface are later replaced by others (Vroman effect, [138]) with a higher affinity for the surface [138,139,140]. The exchange mechanisms are still being explored [139,141,142]. The character of the surface has been shown to affect the affinity [143,144] and the eventual balance between the adsorbed proteins [145,146]. Moreover, the stability of the protein layer on the NP can affect the NP-RBC interaction (adhesion [147], hemolytic activity [49]), and cellular uptake [148]).

In addition, the protein corona composition is sensitive to the NP surface functionalization. For example, Kelpsiene et al. [149] found that aminated NPPS bind a different set of proteins than carboxylated NPPS.

Notably, modern approaches that use artificial intelligence are now being implemented to predict corona composition and help explain the biological compatibility of NPs [150,151,152,153]. Moreover, the method reported by Bun and colleagues [150] successfully predicted cellular recognition (e.g., cellular uptake by macrophages and cytokine release) and nanotoxicity mediated by functional corona proteins.

As we demonstrated, forming a corona around nanomaterials is a complex process, and, for a comprehensive introduction to this topic, we recommend several reviews [128,129,130,131,132] selected from a long list of relevant publications.

## 5. RBC Interaction with Corona-Coated NP

The NP adhesion to the cell surface is critical in determining their interaction level. The inhibition of NP/RBC adhesion may be induced by covering the surface with corona proteins [148,154]. The corona can be created intentionally (by pretreatment of particles) or spontaneously (following the interaction of NP with plasma proteins). As noted above, in the plasma or other body fluids, the NP/RBC interaction is not with the NP itself but with the particle’s corona proteins. The biophysical explanation for the relationship between protein adsorption onto NP surfaces and the NP interaction with red cells is complicated, as many factors, such as the NP and RBC properties and the environment around them, influence this.

In a previous publication [49], we suggested that the protein coating of NPs should decrease their hemolytic activity. To test this hypothesis, we determined the hemolytic activity of uncoated and albumin-treated NPPS (as albumin is an inhibitor of NP/RBC interaction [147]). It was found that at a concentration of 0.05% albumin, the NPPs’ hemolytic capacity is totally inhibited, despite the fact that, at this concentration, only 30–50% of the NP surface is covered with protein [134,155]. Thus, we concluded that the formation of an albumin corona on NPPs leads to a sharp decrease in their hemolytic activity.

Similarly, Yeo et al. [156] found that gold nanorods treated with a serum to form a protein corona on their surface exhibited hemolytic activity of less than 0.2%, with no observable effect on RBC morphology.

Saha et al. [87] considered a more complex issue. They synthesized a class of cationic NPAu with the same core size (~2 nm) but different surface functionalities induced by changing the surface hydrophobicity and determined their hemolytic activity in the presence and absence of plasma proteins. They found a critical synergy between the chemical functions of the NP surface and the protein corona, with corona formation leading to a sharp decrease in the NP hemolytic activity. The presence of plasma proteins prevented the hemolytic activity of both hydrophilic and hydrophobic NPs [87].

## 6. Methods for Assessment of Nanomaterials’ Hemotoxicity

Nano-toxicology is a fast-developing area of nanoscience and nanotechnology. Current studies on the toxic effects of NPs, aiming to identifying the mechanisms of their harmful effects, are carried out in cell culture and animal models [54,55,57,59,157,158,159,160,161,162].

The toxicity of NPPS has received special attention [162,163,164,165]. These particles can be easily synthesized in a wide range of sizes, and their surfaces can be given different functionality [59]. Thus, they are ideally suited as a model for studying the effect of particle surface characteristics on various biological parameters both in vitro and in vivo. Sarma and colleagues [54] have analyzed the cytotoxic and genotoxic potential of NPPS on human peripheral lymphocytes (in vitro), while Loos et al. [59] have summarized information regarding the effect of functionalized (positively and negatively charged) NPPS on macrophages and THP-1 cells (in vitro). These studies indicate that while polystyrene is non-toxic, functionalized nanoparticles may behave differently than bulk material, and surface chemistry plays a critical role in determining the effect of NPPS on various cells.

The toxicity of NPPS was also analyzed in vivo in animal models [159,160,161,162]. Fan et al. [160] observed the accumulation of fluorescent NPPS in various organs of mice after oral ingestion, including in the liver, kidney, spleen, and pancreas. The main mechanism of damage to the internal organs was the impairment of liver function and lipid metabolism. Yasin and colleagues also identified the striking hepatoxicity of NPPS (in a dose-dependent manner) [162] in rats. In addition, a recent in vivo study showed that PSNPs induced reproductive toxicity [161] in mice, caused fetal growth restriction, and significantly impaired cholesterol metabolism in both the mice’s placenta and the fetus [159].

However, the toxicity and risk associated with the use of NPs still need to be understood in their entirety [95]. The development of an adequate experimental strategy for estimating NPs’ toxicity should include the choice between in vitro (cell lines) and in vivo (animal models) methods or a combination of both, as both methods have advantages and disadvantages. The NP toxic effects on individual cell components and tissues are more accessible for in vitro analysis, while in vivo models enable the assessment of NP toxicity for individual organs or the whole organism [163]. It seems more logical to first test NP toxicity on cells, and if toxic effects are clearly demonstrated, this may spare the need for animal testing, in accordance with the global trend of reducing the number of animal experiments [97,98].

The rapid growth of nanomedicine and the development of more and more new NPs make in vivo toxicity tests undesirable on both ethical and financial grounds, creating an urgent need to develop in vitro cell-based assays that accurately predict in vivo toxicity and facilitate safe nanotechnology.

Of all the cell types [164] that can be used to assess the toxicity of nanomaterials, the choice of RBC as a target cell seems to be the most useful. As noted above, irrespective of their use, source, and route of exposure, NPs enter the bloodstream and interact with RBCs, the major cellular component in the circulation (4–5 million RBCs per 1 μL of blood), producing a negative effect on their functionality. As RBCs are well characterized, accessible, and easy to manipulate, they make an excellent candidate for being the target cells for nanotoxicity assessment.

Numerous studies have examined the NP-RBC interaction, focusing on the hemolytic potential of NPs [49,165], suggesting that this is the critical test of NP safety [75,166]. Although hemolysis tests have been conducted with various NPs, comparing results across studies is difficult due to the variability of protocols implemented for particle characterization and hemolysis testing [52].

The American Society for Testing and Materials (ASTM) published (2008) a standard test protocol for the assessment of NPs’ hemolytic properties [166], which determines the percentage of hemoglobin (Hb) released after NP-RBC interaction. The hemolytic assay has proven to be a promising test for surveying nanomaterial toxicity [167] due to its low cost, good reproducibility, and quick results [77]. To date, hemolytic activity has even been demonstrated with therapeutic NPs in vitro [73,168,169] and in vivo [170,171], indicating the potential adverse effects of NPs, which may limit their applications in nanomedicine.

Cho et al. [172] studied the nanotoxicity of a panel of NPs (CeO_2_, TiO_2_, carbon black, SiO_2_, NiO, Co_3_O_4_, Cr_2_O_3_, CuO, and ZnO). The authors compared the acute lung inflammogenicity in a rat model with in vitro toxicity. For in vitro testing, eight different cell-based assays were used, including epithelial cells, monocytic/macrophage cells, human erythrocytes, and combined culture. Cytotoxicity in differentiated peripheral blood mononuclear cells was the most accurate, demonstrating 89% accuracy and 11% false negative results in predicting acute pulmonary inflammation. However, only hemolysis tests demonstrated a 100% match with lung inflammation at all NP concentrations. Other in vitro cellular assays showed a weaker correlation with in vivo inflammatory activity.

An analysis of the related literature supports the finding that NP-induced hemolytic activity can assess in vivo NP toxicity and has been proposed as a critical test in determining NP hemocompatibility [75,77,166,173]. However, despite the attempts to develop a unified protocol to determine NPs’ hemolytic activity, the measurement conditions used by various research groups still differ significantly [52].

For a universal protocol, it is necessary to consider that forming a protein corona around NP inhibits its effective hemolytic activity. In addition, the interaction between a red cell and a nanoparticle in the bloodstream occurs under flow-induced mechanical stress, which can cause RBC deformation [174] and stimulate NP hemolytic activity [47]. Thus, it would be appropriate to test NP hemocompatibility under mechanical stress conditions in a medium supplemented by plasma proteins or in the plasma itself (and not in a buffer, as is customary in many laboratories).

The ability of an NP to change RBC properties can be expressed as an alteration in its functionality and, in its extreme form, as the destruction of the cell [38,175,176]. Therefore, other properties of red cells, such as their aggregability, deformability, and adhesion to EC, should be considered alternative markers to NP hemolytic activity [38,175,176].

All of the mentioned studies demonstrate the protective role of the protein corona formed on the nanomaterial’s surface, improving the NP hemocompatibility and providing promising options for the design of therapeutic nanomaterials without prohibitive toxic effects.

Thus, we can summarize that the NPs’ characteristics and the plasma composition are the dominant factors determining the NPs’ hemocompatibility. Additional factors that can affect the NP hemolytic ability inclue the properties of the RBCs themselves and the presence of mechanical stress (Figure 2). For these reasons, when developing a protocol for testing the hemolytic activity of NPs, it is necessary to consider all four factors.

## 7. Conclusions

The application of nanotechnology to medicine is expected to have a revolutionary impact on health care [115,116,117] and has already stimulated the emergence of relatively new areas, such as nanotoxicology. It is evident that, with the expansion of NP use, the need to assess the toxicity of new materials also grows. However, assessing NP toxicity is a costly process that includes several steps. As discussed above and further detailed in additional publications [70,177,178], an in vitro assay is a superior method for preliminary toxicity assessments. While, at present, the tests for NPs’ hemolytic activity are the most widely adopted, they ignore several important factors, particularly the need to assess hemolysis in the presence of plasma proteins and under conditions of mechanical stress. Furthermore, when considering the use of NPs for treating pathologies related to impaired RBC function (e.g., diabetes, hemoglobinopathies, and others), toxicity testing should be carried out using cells specific to these conditions.

Finally, we hope this review will promote further research on NP-RBC interactions and encourage researchers to develop simple and effective in vitro tests to assess NP hemocompatibility.

## Figures and Tables

**Figure 1 micromachines-13-02091-f001:**
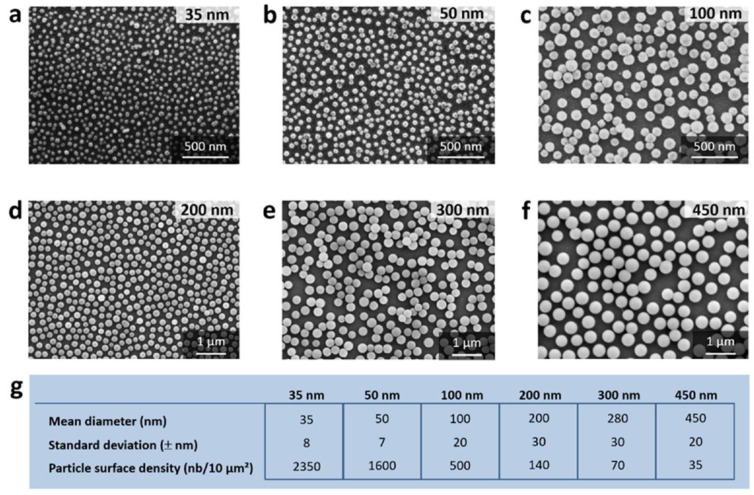
The nanoparticle-based monolayers. (**a**–**f**), scanning electron microscopy images of the different fluorescent silica nanoparticles (NPs) monolayers, constructed with 35 nm, 50 nm, 100 nm, 200 nm, 300, and 450 nm NPs, respectively. (**g**) Table of the mean sizes, standard deviation, and NPs surface density (number of particles per 10 µm^2^) corresponding to each NP size (all these data were obtained using ImageJ with manual thresholding). “Reproduced from [19]”.

**Figure 2 micromachines-13-02091-f002:**
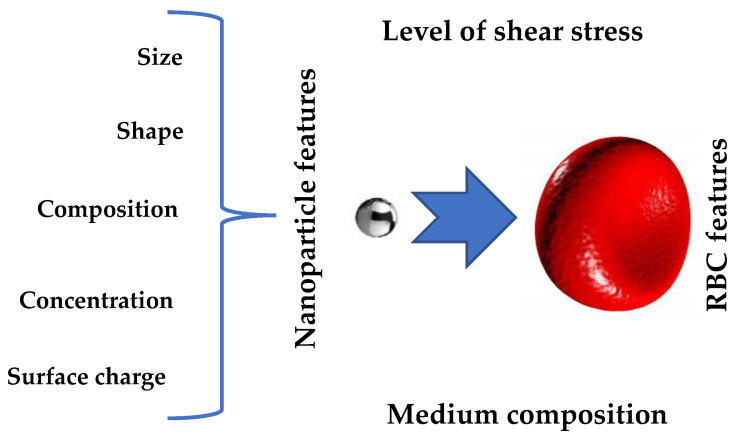
Factors that affect the hemolytic activity of nanoparticles.

**Table 1 micromachines-13-02091-t001:** List of NPs and their hemolytic activity.

#	NPs	Size, nm	[NP] in Blood, mg/mL	Ref.
1	PS plain	50; 100; 200	0.001 ÷ 0.05	[51,52,53,54,55]
2	Amino-modified PS	50; 100; 200	0.001 ÷ 0.05	[56,57,58]
3	Carboxyl-modified PS	50; 100; 200	0.001 ÷ 0.05	[56,59]
4	TiO_2_	15; 20; 30	0.02 ÷ 1.0	[60,61,62]
5	Fe_3_O_4_	10; 20; 50; 100	1.5 ÷ 4.0	[63,64,65]
7	MgO	25; 40; 60	1.0 ÷ 20.0	[66,67]
8	Gold	3; 5; 50; 100	0.05 ÷ 0.5	[68,69,70,71]
9	Silver	35	0.020 ÷ 1.0	[68,72,73,74]
10	Mesoporous hollow silica	60; 110	0.03 ÷ 1.5	[75,76,77]
11	ZnO	20; 50	0.8 ÷ 10	[78,79,80]
12	Selenium	70–200	0.0005 ÷ 0.2	[32,81]

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
