# Peer review of "Hemolytic Activity of Nanoparticles as a Marker of Their Hemocompatibility"

_micromachines, 2022, doi:10.3390/mi13122091_

Round 1

Reviewer 1 Report

The authors address a highly relevant topic in the field of Nanomedicine, since the determination of NPs toxicity is crucial to ensure their safer use in biomedical studies and applications. However, there are significant issues that need to be clarified/corrected by the authors before acceptance of their paper for publication in Micromachines. A list of some of these issues is presented below: 

1. The authors should update their literature review and also cite the most recent studies reporting the evaluation of toxicicity of NPs, especially those describing hemocompatibility studies. 

2. A serious issue in this paper is the lack of figures and schemes. These elements are crucial to enhance any article and must be included. Also, the authors should prepare a table including all discussed the papers plus the ones they did not discuss in the text. These tables are highly useful for a reader that wants to have an overview of the theme. 

3. The authors should carefully review the language and correct all grammar errors before acceptance. 

4. The authors only mention that NPs are sorrounded by a corona of proteins when they enter into the bloodstream. However, the authors should also consider, mention, and discuss the interaction of NPs with other organic molecules (such as hormones, vitamins and sugars) and inorganic ions in such a biological environment.  

5. The authors should carefully review the ext and make it more coherent. Incomplete sentences should be rephrased (e.g. page 3, lines 138-139; page 4, lines 183-185, etc.)

6. "Machine learning method" (page 4, lines 198-199) is a confusing term and should be replaced.

7. The author do not need to use quotation marks when they mention the word corona. 

8. In vivo and in vivo should always be italicized. 

9. Naïve (page 5, line 219) is not an adequate term. It is better to use a different word in this case, such as bare or uncoated. 

10. The work of Saha et al. should be discussed in more detail (page 5, lines 233-239). What were the chemical functionalities investigated by the authors? The same thing applies to the work of Cho et al. What kind of NPs were studied by the authors?

11. The authors should be more careful with the acronyms used. For example, RBCs is sometimes mentioned as such and sometimes mentioned in its entire form, red blood cells. 

11. NPs may not cause hemolysis or morphological changes in RBCs, but they can alter the way hemoglobin binds O2. This is mentioned only briefly (page 3, lines 145-150), but more details and discussions should be given in this regard.

Author Response

Review 1

The authors address a highly relevant topic in the field of Nanomedicine, since the determination of NPs toxicity is crucial to ensure their safer use in biomedical studies and applications. However, there are significant issues that need to be clarified/corrected by the authors before acceptance of their paper for publication in Micromachines. A list of some of these issues is presented below:

We thank the reviewer for his kind support!!

  1. The authors should update their literature review and also cite the most recent studies reporting the evaluation of toxicity of NPs, especially those describing hemocompatibility studies.

We thank the reviewer for the comment; the text was modified accordingly.

  1. A serious issue in this paper is the lack of figures and schemes. These elements are crucial to enhance any article and must be included. Also, the authors should prepare a table including all discussed the papers plus the ones they did not discuss in the text. These tables are highly useful for a reader that wants to have an overview of the theme.

We thank the reviewer for the comment; the text was modified accordingly. We included in the review text the table mentioned by the reviewer and added a diagram describing the main factors affecting the hemolytic activity of nanoparticles.

  1. 3. The authors should carefully review the language and correct all grammar errors before acceptance.

We thank the reviewer for the comment; the text was modified accordingly.

  1. 4. The authors only mention that NPs are sorrounded by a corona of proteins when they enter into the bloodstream. However, the authors should also consider, mention, and discuss the interaction of NPs with other organic molecules (such as hormones, vitamins and sugars) and inorganic ions in such a biological environment.

We thank the reviewer for the comment; the text was modified accordingly (see lines 83-85).

  1. 5. The authors should carefully review the text and make it more coherent. Incomplete sentences should be rephrased (e.g. page 3, lines 138-139; page 4, lines 183-185, etc.)

We thank the reviewer for the comment; the text was modified accordingly (see Lines 193-97).

  1. 6. "Machine learning method" (page 4, lines 198-199) is a confusing term and should be replaced.

We thank the reviewer for the comment; the text was modified accordingly (see Line 205).

  1. 7. The author do not need to use quotation marks when they mention the word corona.

We thank the reviewer for the comment; the text was modified accordingly.

  1. 8. In vivo and in vivo should always be italicized.

We thank the reviewer for the comment; the text was modified accordingly.

  1. 9. Naïve (page 5, line 219) is not an adequate term. It is better to use a different word in this case, such as bare or uncoated.

We thank the reviewer for the comment; the text was modified accordingly see Line 225.

  1. 10. The work of Saha et al. should be discussed in more detail (page 5, lines 233-239). What were the chemical functionalities investigated by the authors? The same thing applies to the work of Cho et al. What kind of NPs were studied by the authors?

We thank the reviewer for the comment; the text was modified accordingly (see Lines 234 -240).

  1. 11. The authors should be more careful with the acronyms used. For example, RBCs is sometimes mentioned as such and sometimes mentioned in its entire form, red blood cells.

We thank the reviewer for the comment; the text was modified.

  1. NPs may not cause hemolysis or morphological changes in RBCs, but they can alter the way hemoglobin binds O2. This is mentioned only briefly (page 3, lines 145-150), but more details and discussions should be given in this regard.

We thank the reviewer for the comment; the text was modified accordingly (see Lines 136-159).

Reviewer 2 Report

The topic of the review is interesting and have impact on the field. However, my suggestion is to reorganize and rewrite the paper. Below are some comments:

1.                Authors should revise the grammar and several phrases must be rewritten. Here, are some examples:

Apart from size, nanomaterials may be classified in terms of their physical parameters, e.g., electrical charge; chemical characteristics, such as the composition of the nanomaterials core or shell; shape (tubes, films, rods, etc.); and origin: natural nanomaterials (in volcanic dust, viral particles, etc.) and artificial nanomaterials”. This is very confuse, authors should rewrite this sentence.

 “The tiny size of nanomaterials allows them to pass more easily through cell membranes and other biological barriers” . This should be rewrite and authors have to cite the name of “other biological barriers”. Please, explain better.

 2.                Nanomaterials can bind and perturb biological molecules in cells, such as DNA, lipids, and proteins. Please, explain how nanomaterials can bind and perturb biological molecules? I think the term “perturb biological molecules” is not correct.

 3.                “Independently from their use, source, and exposure (via oral, respiratory, or dermal routes), nanoparticles enter the bloodstream. They are distributed to various organs, where they are partially metabolized, excreted, or retained. For that reason, recently, a new branch of science named nanotoxicology [17,18], which aims to study the dangerous effects of nanomaterials on human health and the environment, has emerged.” Please, rewrite this sentence and explain better how can nanoparticles enter the bloodstream.

 4.                Recently the mechanisms of NPs toxicity have been studied intensively with [22] or without [23-25] testing the generation of reactive oxygen species (ROS). Several different approaches can be used to assess NP toxicity. These include epidemiological studies, human clinical studies, animal models, and in-vitro tests. Please, explain how ROS can be produced by NPs. I think the phrase “several different approaches…” is not in the correct place.

 5.                The paragraph Lines 78 – 86 must be reorganized and rewritten. There are also several grammar mistakes.

6.                The authors should include images from NPs characterization using the techniques such as SEM, TEM, XDR, FTIR, etc, and include some discussion regarding their different shapes and their hemocompatibility.

7.                The special attention to evaluating the undesirable effects of gold and silver nanoparticles, which are increasingly used in biomedical applications. Why gold and silver nanoparticles have “special attention” compared to other types of NPs?

8.                That was demonstrated that incubation of RBC with silver NP (NPAg), which produced numerous toxic products, specifically ROS [55], caused strong hemolysis. However, the interaction of NPAg with a red cell leads not only to a change its membrane condition but also to an alteration of intracellular hemoglobin properties. Please rewrite this sentence and explain better.

99.   “The interaction of erythrocytes with nanoparticles generally adversely affects the ability of hemoglobin to bind oxygen.” How is it possible?

110.   “Due to RBCs being the most abundant circulating cells, the development of various RBC-based artificial drug delivery systems over the past few decades.” This is confused, please rewrite and complete this phrase.

111.   Using a range of in-vitro toxicity tests, including the hemolytic activity of NP, the authors [110] demonstrated that only the hemolysis assay has 100% consistency with lung inflammation in any dose. Other cell-based in vitro assays showed a poorer correlation with in vivo inflammogenicity [110]. Which doses? And what are the “other cell-based” cited by the paper?

112.   In the conclusion section, authors cited the meaning of nanomedicine. Please, include this information in the introduction.

Author Response

Review 2

We thank the reviewer for his kind support!!

  1. Authors should revise the grammar and several phrases must be rewritten. Here, are some examples:

We thank the reviewer for the comment; the text was modified reviewer's recommendation accordingly.

“Apart from size, nanomaterials may be classified in terms of their physical parameters, e.g., electrical charge; chemical characteristics, such as the composition of the nanomaterials core or shell; shape (tubes, films, rods, etc.); and origin: natural nanomaterials (in volcanic dust, viral particles, etc.) and artificial nanomaterials”. This is very confuse, authors should rewrite this sentence.

We thank the reviewer for the comment; we deleted this paragraph.

 “The tiny size of nanomaterials allows them to pass more easily through cell membranes and other biological barriers” . This should be rewrite and authors have to cite the name of “other biological barriers”. Please, explain better.

We thank the reviewer for the comment; the text was modified accordingly (see Lines 41-46).

  1. Nanomaterials can bind and perturb biological molecules in cells, such as DNA, lipids, and proteins. Please, explain how nanomaterials can bind and perturb biological molecules? I think the term “perturb biological molecules” is not correct.

We thank the reviewer for the comment; the text was modified accordingly (see Lines 82-84).

  1. Independently from their use, source, and exposure (via oral, respiratory, or dermal routes), nanoparticles enter the bloodstream. They are distributed to various organs, where they are partially metabolized, excreted, or retained. For that reason, recently, a new branch of science named nanotoxicology [17,18], which aims to study the dangerous effects of nanomaterials on human health and the environment, has emerged.” Please, rewrite this sentence and explain better how can nanoparticles enter the bloodstream.

We thank the reviewer for the comment; the text was modified accordingly (see Lines 70-80).

  1. Recently the mechanisms of NPs toxicity have been studied intensively with [22] or without [23-25] testing the generation of reactive oxygen species (ROS). Several different approaches can be used to assess NP toxicity. These include epidemiological studies, human clinical studies, animal models, and in-vitro tests. Please, explain how ROS can be produced by NPs. I think the phrase “several different approaches…” is not in the correct place.

We thank the reviewer for the comment; the text was deleted.

  1. The paragraph Lines 78 – 86 must be reorganized and rewritten. There are also several grammar mistakes.

We thank the reviewer for the comment; the text was modified accordingly (see Lines 74-84).

  1. The authors should include images from NPs characterization using the techniques such as SEM, TEM, XDR, FTIR, etc,

We thank the reviewer for the comment. We do not consider it necessary to provide images of nanoparticles in this review; instead, we give a wide list of literature on this issue (see Lines 32 -35).

and include some discussion regarding their different shapes and their hemocompatibility.

We discussed the relationship between NP shape and their hemocompatibility (see Lines 65-67).

  1. The special attention to evaluating the undesirable effects of gold and silver nanoparticles, which are increasingly used in biomedical applications. Why gold and silver nanoparticles have “special attention” compared to other types of NPs?

We thank the reviewer for the comment. We have added the required comments to the text of the revised version of the review (see Lines 143-144).

  1. That was demonstrated that incubation of RBC with silver NP (NPAg), which produced numerous toxic products, specifically ROS [55], caused strong hemolysis. However, the interaction of NPAg with a red cell leads not only to a change its membrane condition but also to an alteration of intracellular hemoglobin properties. Please rewrite this sentence and explain better.

We thank the reviewer for the comment; the text was modified accordingly (see Lines 145-146).

  1. "The interaction of erythrocytes with nanoparticles generally adversely affects the ability of hemoglobin to bind oxygen.” How is it possible?

We thank the reviewer for the comment; the text was modified accordingly (see Lines 147-159).

110        " Due to RBCs being the most abundant circulating cells, the development of various RBC-based artificial drug delivery systems over the past few decades.” This is confused, please rewrite and complete this phrase.

We thank the reviewer for the comment; we rewrite the phrase (see Lines 161-162).

  1.   Using a range of in-vitro toxicity tests, including the hemolytic activity of NP, the authors [110] demonstrated that only the hemolysis assay has 100% consistency with lung inflammation in any dose. Other cell-based in vitro assays showed a poorer correlation with in vivo inflammogenicity [110]. Which doses? And what are the “other cell-based” cited by the paper?

We thank the reviewer for the comment; we have written this paragraph (see Lines 294-297).

  1. In the conclusion section, authors cited the meaning of nanomedicine. Please, include this information in the introduction.

We thank the reviewer for the comment; we deleted the concept "nanomedicine" in the conclusion section (see Line 340).

Reviewer 3 Report

This paper reviews an important yet often overlooked aspect of drug delivery systems - interaction with RBC and therefore, represents a timely and much needed advancement of the analysis of this topic (PMID 30149885). I have few minor suggestions for inclusion additional information and references, in order to further boost the impact of this important paper.

  1. Effect of modification on biocompatibility of RBC must be addressed rigorously in vitro and in vivo, comparing side-by-side with control naïve RBCs for sensitivity of the carrier RBC to osmotic lysis, complement, mechanical stress and oxidative stress, as described and in vivo by direct tracing of blood clearance and uptake by RES, lungs, kidneys (PMID 27003833; 29371620).
  2. In addition to helping address the key safety issues of agents, understanding of the injurious host defense reactions such as opsonization, lysis and phagocytosis of micro- and nanoscale objects is necessary to avoid unintended and harmful side effects (PMID  34910334; 33788581; 27856317). This review, therefore, pertains also to RBC-based drug delivery systems (DDSs).
  3. RBC-based DDSs are explored for decades (PMID 34597748; 20410503; 15525799; 3462715; 25151978; 25992439; 8867894). Multiple animal and human studies showed feasibility of drug encapsulation into the RBC (PMID 24968029; 22833997; 22230036; 2128736) and coupling to RBC surface (PMID 32397513; 29365311; 27836986; 26228773; 19616049; 27707522). 
  4. Studies of biocompatibility of RBC modified to attach pharmacological agents, cross-linking compounds, particles and targeting ligands come back almost half a century. It would be great to mention these early studies, such as an important paradigm for modification of RBC for clearance of pathogenic agents using dual-targeted complexes binding these harmful agents to RBC CR1, a privileged site on human RBC that allows macrophages to take “garbage” without damage to RBC carrier (PMID 16735601; 32885023; 24184879; 15728520; 9028340).
  5. Also, of high interest is the effect of streptavidin binding to biotinylated RBCs, leading to inhibition of DAF and CD59, activation of homologous complement system via the alternative pathway, fixation of C3b, formation of the active C3-convertase on the surface of RBC leading to its hemolysis and elimination (PMID 8921172; 8756393; 7537958; 7695090; 1616915; 1824256; 2009954). These studies yielded methodology for coupling cargoes to RBC avoiding RBC lysis by complement, such as using phospholipid anchors instead of modification of RBC membrane glycoproteins (PMID 8440366; 8429223; 8452230; 1991038).
  6. Unintended re-arrangements in the RBC membrane, such as making it more rigid (PMID 34233380) may complicate design of RBC-hitchhiking strategies, which are hot topic today (PMID  35780495 ; 35710322; 35266686; 29992966).

Author Response

Review 3

This paper reviews an important yet often overlooked aspect of drug delivery systems - interaction with RBC and therefore, represents a timely and much needed advancement of the analysis of this topic (PMID 30149885). I have few minor suggestions for inclusion additional information and references, in order to further boost the impact of this important paper.

We thank the reviewer for his kind support!! We thank the reviewer for his recommendation; in the revised version of the review, we cited recommended paper (see Line 30).

Effect of modification on biocompatibility of RBC must be addressed rigorously in vitro and in vivo, comparing side-by-side with control naïve RBCs for sensitivity of the carrier RBC to osmotic lysis, complement, mechanical stress and oxidative stress, as described and in vivo by direct tracing of blood clearance and uptake by RES, lungs, kidneys (PMID 27003833; 29371620).

We thank the reviewer for the comment; the text was modified accordingly (see Lines 170-171).

In addition to helping address the key safety issues of agents, understanding of the injurious host defense reactions such as opsonization, lysis and phagocytosis of micro- and nanoscale objects is necessary to avoid unintended and harmful side effects (PMID  34910334; 33788581; 27856317). This review, therefore, pertains also to RBC-based drug delivery systems (DDSs).

We thank the reviewer for the comment. We took the reviewer's advice and expanded the section " RBC-carrier of nanoparticles "(see Lines 172-173).

RBC-based DDSs are explored for decades (PMID 34597748; 20410503; 15525799; 3462715; 25151978; 25992439; 8867894). Multiple animal and human studies showed feasibility of drug encapsulation into the RBC (PMID 24968029; 22833997; 22230036; 2128736) and coupling to RBC surface (PMID 32397513; 29365311; 27836986; 26228773; 19616049; 27707522).

We thank the reviewer for the comment. We have added several publications from the list, kindly provided by the reviewer. (see Lines 168-169).

Studies of biocompatibility of RBC modified to attach pharmacological agents, cross-linking compounds, particles and targeting ligands come back almost half a century. It would be great to mention these early studies, such as an important paradigm for modification of RBC for clearance of pathogenic agents using dual-targeted complexes binding these harmful agents to RBC CR1, a privileged site on human RBC that allows macrophages to take “garbage” without damage to RBC carrier (PMID 16735601; 32885023; 24184879; 15728520; 9028340).

We thank the reviewer for the comment. We have carefully read the reviewer's recommendation and are forced to reject it. Our review focused on the interaction of RBCs with NPs, and the reviewer's proposal goes beyond this.

Also, of high interest is the effect of streptavidin binding to biotinylated RBCs, leading to inhibition of DAF and CD59, activation of homologous complement system via the alternative pathway, fixation of C3b, formation of the active C3-convertase on the surface of RBC leading to its hemolysis and elimination (PMID 8921172; 8756393; 7537958; 7695090; 1616915; 1824256; 2009954). These studies yielded methodology for coupling cargoes to RBC avoiding RBC lysis by complement, such as using phospholipid anchors instead of modification of RBC membrane glycoproteins (PMID 8440366; 8429223; 8452230; 1991038).

We thank the reviewer for the comment. We have carefully read the reviewer's recommendation and are forced to reject it. Our review focused on the interaction of RBCs with NPs, and the reviewer's proposal goes beyond this.

Unintended re-arrangements in the RBC membrane, such as making it more rigid (PMID 34233380) may complicate design of RBC-hitchhiking strategies, which are hot topic today (PMID  35780495; 35710322; 35266686; 29992966).

We thank the reviewer for the comment. We have carefully read the reviewer's recommendation and are forced to reject it. Our review focused on the interaction of RBCs with NPs, and the reviewer's proposal goes beyond this.

Round 2

Reviewer 1 Report

The authors have made important changes in the text, which enhanced and improved their paper. However, some issues still remain: 

1. This is a review paper; so, Figures reporting interesting results from the literature should be included (e.g. TEM and SEM images of cells and nanoparticles and other relevant results, such as fluorescence microcopy and related studies). This will significantly enhance the paper. 

2. Machine learning should be defined. 

3. The text should be further reviewed and all typos/minor language problems should be corrected.

Author Response

Reviewer 1

The authors have made important changes in the text, which enhanced and improved their paper.

We thank the reviewer for his kind support!!

However, some issues still remain: 

  1. This is a review paper; so, Figures reporting interesting results from the literature should be included (e.g. TEM and SEM images of cells and nanoparticles and other relevant results, such as fluorescence microcopy and related studies). This will significantly enhance the paper. 

We thank the reviewer for the comment; we added relevant image to the manuscript (Fig 1).

  1. Machine learning should be defined. 

We thank the reviewer for the comment; we delated mentioned concept from the revised text (see Line 219).

  1. The text should be further reviewed and all typos/minor language problems should be corrected.

We thank the reviewer for the comment; we corrected the text.

Reviewer 2 Report

The authors made the corrections mentioned in the revision. However, I have some other corrections that is necessary before the acceptance of this paper.

1.      Please, remove the word “their” (Line 78).

2.      Lines 244-245 the authors cited several references. Please, describe some of them on this paragraph.

3.    The authors should include images from NPs characterization using the techniques such as SEM, TEM, XDR, FTIR, etc, and include some discussion regarding their different shapes and their hemocompatibility. I strongly recommend the authors include images to illustrate nanoparticles

4.      The special attention to evaluating the undesirable effects of gold and silver nanoparticles, which are increasingly used in biomedical applications. Why gold and silver nanoparticles have “special attention” compared to other types of NPs? This is not clear.

5.      The authors listed some works in Table 1 however authors should also include recently papers.

Author Response

The authors made the corrections mentioned in the revision. However, I have some other corrections that is necessary before the acceptance of this paper.

  1. Please, remove the word “their” (Line 78).

We thank the reviewer for the comment; we have changed this paragraph. (see Line 81-83).

  1. Lines 244-245 the authors cited several references. Please, describe some of them on this paragraph.

We thank the reviewer for the comment; we added relevant paragraph to the text (see Line 258-76).

  1.   The authors should include images from NPs characterization using the techniques such as SEM, TEM, XDR, FTIR, etc, and include some discussion regarding their different shapes and their hemocompatibility. I strongly recommend the authors include images to illustrate nanoparticles

We thank the reviewer for the comment; we added relevant illustration (see Figure 1).

  1. The special attention to evaluating the undesirable effects of gold and silver nanoparticles, which are increasingly used in biomedical applications. Why gold and silver nanoparticles have “special attention” compared to other types of NPs? This is not clear.

We thank the reviewer for the comment; we added relevant paragraph in to the text (see Line 149-57).

  1. The authors listed some works in Table 1 however authors should also include recently papers.

We thank the reviewer for the comment; we added relevant citations.

Round 3

Reviewer 1 Report

The current version can be accepted for publication. 

Reviewer 2 Report

The authors made the corrections and the paper can be accepted.